# AMP: Automatically Finding Model Parallel Strategies with Heterogeneity Awareness

**Dacheng Li**[c] , **Hongyi Wang**[c] , **Eric Xing**[mcp], **Hao Zhang**[b]

[c] Carnegie Mellon University   [m] Mohamed Bin Zayed University of Artificial Intelligence
[p] Petuum Inc.  [b] University of California, Berkeley

## Abstract

Scaling up model sizes can lead to fundamentally new capabilities in many machine learning (ML) tasks. However, training big models requires strong distributed system expertise to carefully design model-parallel execution strategies that suit the model architectures and cluster setups. In this paper, we develop AMP, a framework that automatically derives such strategies. AMP identifies a valid space of model parallelism strategies and efficiently searches the space for high-performed strategies, by leveraging a cost model designed to capture the heterogeneity of the model and cluster specifications. Unlike existing methods, AMP is specifically tailored to support complex models composed of uneven layers and cluster setups with more heterogeneous accelerators and bandwidth. We evaluate AMP on popular models and cluster setups from public clouds and show that AMP returns parallel strategies that match the expert-tuned strategies on typical cluster setups. On heterogeneous clusters or models with heterogeneous architectures, AMP finds strategies with $1.54\times$ and $1.77\times$ higher throughput than state-of-the-art model-parallel systems, respectively. [1]

## 1 Introduction

Recent progress in language understanding [6, 4], multimodal learning [1], and many other ML applications can largely credit to the use of extremely big models. For example, the largest language models [4, 14] with billions of parameters are found to have fundamentally new capabilities. Due to the increased model size, training such models, however, calls for *model parallelism* execution strategies in order to place the gigantic model computation across multiple, heterogeneous accelerator devices (e.g., TPUs and GPUs), and ensure efficient distributed execution.

Different from typical *data parallelism* strategies [28, 5, 15], model parallelism considers a much larger space of parallelization techniques. In particular, when a model is too large to fit in a single device memory, we can either place the computation of different layers across different devices (a.k.a. *pipeline parallelism*), or partition the computation of particular layers and dispatch parts onto parallel devices (i.e., *tensor model parallelism (TMP)* [23]). Due to their distinct communication and memory requirements, these model parallelism dimensions have different applicable scenarios, which may change with model architectures and cluster setups. For example, TMP results in high communication volumes, hence is favored on devices connected with high bandwidth, but may become less effective with limited communication bandwidth. Designing performant execution strategies requires combining these parallelism dimensions to trade-off among memory, communication, and computation, then tuning the configurations in each parallelism dimension to align with the heterogeneity of the model and the training cluster.

---

[1]Codes and experiment logs are available at `https://github.com/MccRee177/AMP` for reproducibility.

36th Conference on Neural Information Processing Systems (NeurIPS 2022).

Existing model-parallel training strategies are either manually designed by domain experts for one or two specific models, or automatically generated with strong assumptions on the model architecture or cluster topologies. For example, the *3D parallelism* strategy adopted in Megatron-LM [23] and DeepSpeed [21] is manually specialized for transformer-based language models, assuming a fixed cluster setup and the model having the same layer repeated [23, 21, 19]. Therefore, they can hardly generalize to models with diverse layer compositions (i.e., *heterogeneous models*), or clusters with mixed types of accelerators and network switches (i.e., *heterogeneous clusters*). This motivates the core questions in this paper: *Is it possible to automatically find the best model-parallel strategies for more heterogeneous models and clusters?*

Given an arbitrary cluster and model setup, finding highly performed model-parallel strategies is difficult for the following reasons. First, when the model has diverse types of layers, existing heuristics on deciding the assignment of layers to pipeline stages (and balance stage workloads) can be invalidated, because different layers exhibit distinct execution and communication costs. Second, evaluating a model-parallel strategy on a large cluster can be expensive, and prevents search-based methods [11] that depend on extensive real evaluations. However, existing cost models do not capture the heterogeneity in the model or cluster (e.g. [24] assumes a uniform bandwidth in the cost model). In this paper, we present solutions to these challenges. To address the layer-stage assignment problem, we develop a new dynamic programming algorithm running in polynomial time. To evaluate a large number of strategies within a limited budget, we develop a new cost model which serves as a cheaper proxy to evaluate model-parallel strategies. We name our method **AMP**. In summary, AMP makes the following contributions:

- We identify factors that influence model-parallel performance in heterogeneous settings, and define strategies based on these factors.

- We present a new cost model that handles heterogeneity in both the model and the cluster to evaluate the quality of model parallel strategies with a minimal amount of real trials.

- We develop a dynamic programming approach to handle the load imbalance issue from heterogeneous models in the pipeline layer assignment problem.

- We empirically show that existing systems output sub-optimal strategies in heterogeneous settings. In contrast, AMP achieves $1.54\times$ speedup with heterogeneous clusters, and $1.77\times$ speedup on heterogeneous model architectures compared to state-of-the-art heuristics.

## 2 Background and Related Work

In this section, we discuss the trade-offs of existing model-parallel strategies, and the need for composing them with the awareness of model and cluster heterogeneity.

**Data-parallel (DP) strategies.** DP partitions the input data batch evenly among workers, and each worker holds an entire model replica [15, 22, 9]. At each iteration, each worker computes gradients over its assigned batch; the gradients are then synchronized among workers before the next iteration. DP requires each worker to hold an entire model replica, hence cannot be directly used to train models with massive parameters.

**Tensor model-parallel (TMP) strategies.** TMP, proposed by Megatron-LM [23], is a popular model-parallel approach for large transformer models. In TMP, the layer weights of each two consecutive layers are partitioned row-wise (*i.e.,* input dimension) first, then column-wise (*i.e.,* output dimension) [23]. TMP removes the need for synchronizing the intermediate output of the very first layer, but requires heavy cross-device communications afterward. TMP is normally combined with data parallelism to increase the training throughput [18].

**Pipeline-parallel (PP) strategies.** In PP, layers are placed across GPUs, the training mini-batch is split into smaller micro-batches. The forward and backward computation is then pipelined across micro-batches. PP requires less communication than DP and TMP, but suffers from device idle (*i.e.,* pipeline bubbles) [10, 18]. Despite synchronous pipelining schedules such as GPipe [10], PipeDream [18] proposes an 1F1B asynchronous schedule to reduce the pipeline bubble. TeraPipe develops token-level pipelining schedules in a single training sequence for auto-regressive models [16].

**Automatic partition.**    Prior work, *e.g.,* FlexFlow, PipeDream, and DAPPLE explore automatic computation graph partitioning over a few typical cluster device setups [11, 18, 7], such as NVIDIA DGX workstations connected with Infiband switches. For partitioning the model over multiple devices, the aforementioned work leverages carefully designed cost models to select the desired strategy over the search space.

**3D parallelism.**    To trade-off the scalability, memory footprint, and device utilization, DP, MP, and PP have to be used together (which is also known as *3D parallelism*). DeepSpeed carefully designs a 3D parallelism strategy to train a massive-scale language model with 17 billion parameters [21]. [19] manually tunes the degrees of DP, MP, PP to scale to 3072 GPUs – both of them are *not automatic*. The closest to our work is Piper [24], which leverages a cost model and a dynamic programming algorithm to search over the 3D strategy space. However, Piper works only with transformer-based models, and assumes the same type of devices in the cluster are homogeneously connected with equal bandwidth, ignoring heterogeneous setups. In contrast, AMP is designed to capture the heterogeneity of the given ML model and the deployed cluster, reflected in both the search space design and the searching algorithm (§ 3).

**Heterogeneity in models and clusters.**    Newer generations of GPU devices and Ethernet switches emerge rapidly, thus it is common nowadays that multiple types of devices are deployed together in a training cluster. Conducting distributed training over such types of heterogeneous clusters introduces extra challenges. For instance, a poor load balancing strategy can cause straggler effects easily; heterogeneous memory capacity among GPUs introduces new constraints on model and data partitioning. Apart from hardware heterogeneity, DL models also introduce another dimension of heterogeneity. Many today's popular models are composed with diverse layers with different sizes and computing characteristics, *e.g.,* typical convolution networks have their layer width growing with depth [8], which would require distinct partitioning and placement strategies for each layer during model parallelism.

## 3   Methods

### 3.1   Problem Formulation

Formally, the inputs of AMP are (i) the model, (ii) the cluster, and (iii) the global batch size ($gbs$). A deep learning model $\mathcal{W}$ is represented as a graph, *i.e.,* $\mathcal{W} = (C_{\mathcal{W}}, V_{\mathcal{W}})$. $C_{\mathcal{W}}$ is a set of vertices where each vertex $c_{\mathcal{W},i} \in C_{\mathcal{W}}, i \in \{i, \cdots, |C_{\mathcal{W}}|\}$ represents a layer, and each edge $v_{\mathcal{W},i} = (c_{\mathcal{W},i}; c_{\mathcal{W},i+1}) \in V_{\mathcal{W}}$ represents the tensor (activation) between layer $c_{\mathcal{W},i}$ and layer $c_{\mathcal{W},i+1}$. Additionally, we denote the *execution cost* of a layer $c_{\mathcal{W},i}$ as $c_i$, and the *communication volume* of an edge $v_{\mathcal{W},i}$ as $v_i$. A cluster is represented as $\mathcal{C} = \{\mathcal{D}, B\}$ where $\mathcal{D} = \{d_i | 1 \leq i \leq |\mathcal{D}|\}$ denotes a set of devices. $B \in \mathcal{R}^{|\mathcal{D}| \times |\mathcal{D}|}$ is a symmetric matrix where each matrix element $b_{ij}$ represents bandwidth between device $d_i$ and $d_j$. The goal of AMP is to output a high throughput parallel training strategy $s$ given the hardware and model configuration. $\mathcal{S}$ denotes the entire space of candidate strategies. We formulate this as an optimization problem:

$$s^{\star} = \arg\min_{s \in \mathcal{S}} f(s; \mathcal{W}, \mathcal{C}, gbs) \qquad (1)$$

where $f(\cdot)$ denotes per iteration running time as a function of the strategy $s$, conditioned on the given ML model, cluster, and user-specified global batch size.

### 3.2   Method Overview

To solve the optimization problem in Equation 1, we identify factors that influence the system performance (§3.3). However, there are two major difficulty in solving this optimization with these identified factors. First is how to evaluate $f(\cdot)$. One possible solution is to launch real trials, but this is prohibitively expensive because the entire optimization procedure may require evaluating a large number of strategies. Rather we develop a cost model (§ 3.4) that can estimate $f(\cdot)$ quickly. Second, the discrete nature of the optimization problem disallows us to leverage gradient-based optimization methods. However, evaluating all strategies in the search space $\mathcal{S}$ is slow even with the cost model. One way to efficiently optimize the objective is to exploit structure in the strategy

space $\mathcal{S}$, and prune areas with no promising candidates. Using this principle, we develop a dynamic programming approach (§3.5), which prunes out candidates with worse performance before invoking the cost model.

## 3.3 Representation of a Strategy

At a high level, a strategy $s$ defines how to train model $\{C_\mathcal{W}, V_\mathcal{W}\}$ on a given cluster $\{\mathcal{D}, B\}$ with the batch size $gbs$. Concretely, a strategy $s$ that composes DP, TMP, and PP strategies can be represented by a few key aspects, *e.g.,* how many model replicas should be created (*i.e.,* how many pieces should a mini-batch be evenly split) in DP; how to partition model layers (for PP) or layer parameters (for TMP); and how shall the global batch size decompose into micro-batches in PP semantics. In AMP, we represent a strategy $s$ with the following four aspects:

**1. Parallelism degrees.** Since PP, DP, and TMP have different characteristics, different degrees of each affect the system performance. We denote the degrees to be $pp$, $dp$, $tmp$. The tuple $(pp, dp, tmp)$ defines a three-dimensional grid, which satisfies the constraint $pp \times dp \times tmp = |\mathcal{D}|; pp, dp, tmp \in \{1, \ldots, |\mathcal{D}|\}$. This tuple generates $pp \times dp \times tmp$ virtual ranks that carry out corresponding tasks. For instance, ranks in the same data parallelism group communicate with each other to finish the gradient update. These ranks are bijectively mapped to available devices $\mathcal{D}$ for execution.

**2. Device placement.** In a heterogeneous environment, physical devices can have different connection bandwidth to each other, thus placing physical devices in different virtual ranks leads to different performance. For instance, if ranks in a communication-heavy group are all allocated with physical devices with high bandwidth, the per iteration time will be shorter. Formally, We denote a device placement as a bijective function from virtual ranks to devices $p : \{1, \ldots, pp\} \times \{1, \ldots, dp\} \times \{1, \ldots, tmp\} \mapsto \mathcal{D}$.

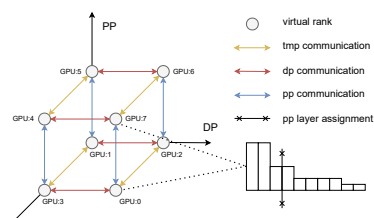

**3. Micro-batch size.** Micro-batch size ($mbs$) is the batch size a device proceeds at a time in the pipeline parallelism execution (§2). When $mbs$ varies, devices usually have different utilization [18, 26]. For instance, a larger micro-batch size usually leads to better GPU utilization because the GPU can better parallel the computation. However, a larger micro-batch size also leads to a larger pipeline bubble, which makes selecting $mbs$ hard.

Figure 1: An example strategy with parallelism degrees $= (2, 2, 2)$. Note that there are 8! possible GPU assignments to these 8 virtual ranks. Assuming a uniform bandwidth, a uniform-layer assignment cuts at the 5th layer, while a more balanced assignment cuts at the 3rd layer.

**4. Pipeline layer assignment.** Different assignments of layers to each device can also affect the system throughput. On the one hand, when activation volume $v$ is large in the boundary, the next device needs to wait longer, resulting in a longer per iteration time. On the other hand, when a device is assigned a much higher load than other devices, it blocks all other devices and thus incurs a longer iteration time. We denote a feasible pipeline layer assignment as a function $a : C_\mathcal{W} \mapsto R$ that maps layers to the stage index $(1, \ldots, pp)$ in a pipeline.

In summary, a strategy $s$ can be defined by a 6-dimensional tuple:

$$\mathcal{S} \triangleq (pp, dp, tmp, p, mbs, a) \tag{2}$$

## 3.4 Cost Model

Since launching real trials to obtain the actual $f(\cdot)$ is expensive, we develop a cost model that serves as a fast estimation of $f(\cdot)$. In this section, we introduce the proposed cost model.

**Communication primitive.** We consider the ring topology for all-reduce collective communication in both DP and TMP [22]. Assume that the whole message size is $M$. In a communication group of $n$ workers with uniform bandwidth $B$, the communication time of a single ring all-reduce operation can be modeled as [13]:

$$T_{allreduce} = \frac{2(n-1)M}{nB} \tag{3}$$

In a group with non-uniform bandwidth, we approximate the bandwidth with the lowest bandwidth $B'$. For point-to-point (P2P) communication operations between consecutive pipeline stages, we

| Variable | Meaning |
|---|---|
| $\mathcal{D}$ | set of devices in the cluster |
| $M$ | message size in an all-reduce operation |
| $n$ | number of workers in an all-reduce operation |
| $B$ | bandwidth between workers in an all-reduce operation |
| $v$ | activation volume for a P2P operation |
| $b$ | bandwidth for a P2P operation |
| $k$ | number of stages in a pipeline |
| $e_i$ | communication time for the P2P operation between the $i^{th}$ and $(i+1)^{th}$ stage |
| $t_i$ | execution time of the $i^{th}$ stage in a pipeline |
| $T_i^{pp}$ | execution time of the $i^{th}$ pipeline |
| $t_j^{layer}$ | execution time of a single layer $j$ |
| $gas$ | number of micro-batches in a pipeline |
| $L$ | number of layers in $\mathcal{W}$ |

Table 1: Major variables and their meanings in the cost model.

assume the communication operation can use the full bandwidth: $e = \frac{v}{b}$ [11]. Here $v$ is the activation volume and $b$ is the bandwidth between these two GPUs.

**Top-down run time decomposition.** We now present our cost model designed from a *top-down* view. First, DP is at the **top** granularity, where there are $dp$ pipelines running in parallel. It synchronizes gradients when all pipelines finish their schedules. It does not consider how each pipeline is executed. At the DP level, the system run-time can be decomposed into two terms:

$$T = \max\{T_i^{pp}|1 \le i \le dp\} + \max_{d \in \mathcal{D}} T_d^{dpsync} \tag{4}$$

$T_i^{pp}$ is the time taken to complete a single pipeline $i$. $T_d^{dpsync}$ denotes the time for a device $d$ to synchronize gradients with an all-reduce operation. We model the latter term by using Equation 3 with the message size equal to the number of model parameters held at device d. Since different pipelines may finish with different speeds, and different devices may finish their data parallelism synchronization with different speeds, we take maximum over these two terms respectively.

At a **lower** granularity, each pipeline executes a training schedule that includes activation/gradients exchange at a layer level. It does not need to consider how each layer is executed, which is carried out by MP workers. Assume the number of stages in a pipeline is $k$, and the communication time between the $i^{th}$ and $(i+1)^{th}$ is $e_i$, and the execution time for the $i^{th}$ stage is $t_i$. We model $t_i$ as the sum over execution time $t_j^{layer}$ in the stage for layer j in the stage: $t_i = \sum_{a(j)=i} t_j^{layer}$. We model the execution time for a single pipeline as [2]:

$$T^{pp} = (gas - 1) \cdot \max\{t_i|1 \le i \le pp\} + \sum_{i=1}^{pp-1} e_i + \sum_{i=1}^{pp} t_i \tag{5}$$

Intuitively, the first term captures the straggler effect, which gets amplified by the number of micro-batches $gas$. If we assume $gas = 1$, Equation 5 reduces to $T^{pp} = \sum_{i=1}^{pp-1} e_i + \sum_{i=1}^{pp} t_i$. This aligns with the actual schedule: the mini-batch simply performs a single forward pass and a single backward pass, and sends or receives gradients or activations. If we assume all layers have the same execution cost and the cluster is equipped with the same bandwidth, the optimal solution is simply to balance the number of layers. This heuristic is often adopted as the default layer assignment solution in current systems[23].

At the **bottom** granularity, MP workers consider how a single layer is executed. The run-time of a single layer $t_j^{layer}$ is composed of the layer computation time (*e.g.,* Matrix multiplication) and all-reduce communication time. One way to estimate $t_j^{layer}$ is using the number of floating point operations (FLOPs) for computation time, and Equation 3 for communication time. However, we find this inaccurate in our setting. Concretely, when $mp$ grows, the arithmetic intensity of each matrix multiplication becomes lower. In such scenarios, a matrix multiplication with only a few FLOPs can take a long time to execute. Thus, we simply obtain $t_j^{layer}$ by profiling several tensor model parallelism settings and store results in a lookup table. This profiling is feasible because it only needs to be done once in the entire optimization.

---

[2] We describe how we consider overlapping in the P2P communication and computation in §6.1.

| Cluster Size | 2×2 | 4×4 | 8×8 | 16×16 |
|---|---|---|---|---|
| Optimization time (seconds) | 58.0 | 325.6 | 890.3 | 1627.4 |

Table 2: AMP optimization time with respect to the cluster size. Results are obtained with a 24-layer GPT-2 model and a global batch size 32.

## 3.5 Pipeline Layer Assignment

A key observation in the layer assignment problem is that only *consecutive* layers can be assigned to the same stage. In particular, computation up to layer $i$ can be reused for that of layer $i + 1$. This pattern fits neatly in a dynamic programming solution. To handle the max term in objective 5, we use an auxiliary variable tolerance $m$. $m$ takes value from all possible execution time for a single stage. For instance, it can be the sum of the execution time of the first and the third layer because they can form a valid pipeline stage. However, it can not be that of the first and the third layer because they are not consecutive. Using $m$, we rewrite an objective for the dynamic programming algorithm:

$$(gas - 1) \cdot \max\{0, \max\{t_i | 1 \leq i \leq pp\} - m\} + \sum_{i=1}^{pp-1} e_i + \sum_{i=1}^{pp} t_i \qquad (6)$$

Note that when $m = 0$, Equation 6 reduces to the desired objective 5. Formally, we store the optimal assignments and the corresponding cost up to layer $i$ with $j$ stages and variable $m$ in table $dp[i][j][m]$. Let $L$ be the number of layers, then $dp[L][k][0]$ stores the solution. We induct on the number of stages in the pipeline.

**Base case ($j = 1$).** The only possible assignment assigns all $i$ layers into a single stage. Denote the sum of execution time for these $i$ layers as $t_1$. The associated optimal cost is thus

$$g(i) = (gas - 1) \times \max\{0, t_1 - m\} + t_1 \qquad (7)$$

We store the table entry $dp[i][0][m]$ with $(i, g(i))$ for all possible $i$ and $m$.

**Recursive step.** To compute $dp[i][j][m]$ when $j > 1$, we enumerate all possible positions of the last cut $i'$ for all $i' < i$. Denote the execution time from layer $i'$ to layer $i$ as $t_{2,i'}$, let the value stored at $dp[i'][j-1][\max(t_{2,i'}, m)]$ be $(a_{i'}, c_{i'})$. Denote the longest stage time in $a_{i'}$ as $t_{1,i'}$, the communication cost between the $(j-1)^{th}$ and $j^{th}$ device at layer $i'$ as $e_{i',j}$. The cost associated with the last cut at $i'$ is: [3]

$$g(i') = c_{i'} + (gas - 1) \times \max\{0, t_{2,i'} - m\} + t_{2,i'} + e_{i',j} \qquad (8)$$

Select the optimal last cut position $i'_{opt} = \operatorname{argmin}_{i' < i} g(i')$. The dynamic programming table entry $dp[i][j][m]$ is updated as: $(a_{i'_{opt}} \cup (i - i'_{opt}), g(i'_{opt}))$.

**Running time.** There are $\sum_{i=1}^{L} i = \mathcal{O}(L^2)$ possible values for $m$. Thus, our problem size is $\mathcal{O}(L \times k \times L^2)$, each sub-problem takes $\mathcal{O}(L)$ time. Thus the total running time for the dynamic programming algorithm is $\mathcal{O}(kL^4)$, while a brute force solution takes $\mathcal{O}\binom{L-1}{K-1}$.

## 3.6 Optimization Procedure

In this section, we introduce the overall optimization procedure to find the best strategies defined at § 3.3. First, we enumerate all possible parallelism degrees and micro-batch sizes. For each parallelism degree and micro-batch size, we deterministically solve for the optimal pipeline layer assignment using the algorithm at § 3.5. We adopt the heuristics at [19] to solve the device placement problem: prioritize placing model parallelism workers in the same node. If there is more available intra-node bandwidth, then place data parallelism workers and pipeline parallelism workers. The procedure is written in Algorithm 1. More details can be found in § 6.2

---

[3]proof can be found in § 6.4

**Algorithm 1:** Optimization procedure

**Input:** $\mathcal{C}$, $\mathcal{W}$, $gbs$, budget

1   degrees = enumerate_degrees($\mathcal{C}$)
2   record = set()
3   **for** *d in degrees* **do**
      /* Possible micro-batch size given data parallel degree            */
4      **for** *mbs in enumerate_(d.dp)* **do**
5         $p$ = placement($\mathcal{C}, d$) // Device placement heuristics
6         $a$ = pipe_ast($\mathcal{W}, d.pp, p$) // Optimal pipeline assignment
7         $s = (a, mbs, d, p)$ // current strategy
8         cost = estimate($s$) // our cost model
9         record.add($s$, cost)

    /* run top predicted strategies                                              */
10   best_s = run(record, budget)
11   **return** best_s

|  | Megatron | AMP | SA |
|---|---|---|---|
| Min | 1.32 | **1.20** | 1.57 |
| Mean | $2.41 \pm 1.29$ | $\mathbf{1.43 \pm 0.18}$ | $1.82 \pm 0.17$ |

Table 3: Best and average strategies under homogeneous setup over Top 10 candidates (scale: seconds)

**Complexity of the overall optimization.** The number of factors of an integer $N$ can be loosely upper bounded by $\mathcal{O}(N^{1/2})$. Thus, the number of possible degrees can be upper bounded by $\mathcal{O}(|\mathcal{D}|^{1/2})) \times \mathcal{O}(|\mathcal{D}|^{1/2}) = \mathcal{O}(|\mathcal{D}|)$, and the number of possible micro-batch sizes can be upper bounded by $\mathcal{O}(gbs^{1/2})$. The pipeline assignment algorithm run-time is bounded by $\mathcal{O}(kL^4) = \mathcal{O}(|\mathcal{D}| \times L^4)$ (section 3.5). Thus, the total run time can be upper bounded by $\mathcal{O}(gbs^{1/2} \times |\mathcal{D}| \times |\mathcal{D}| \times L^4) = (gbs^{1/2} \times |\mathcal{D}|^2 \times L^4)$. Empirically, we verify the optimization time with respect to the cluster size in Table 2. We find that the actual optimization time is near linear with respect to the cluster size.

## 4   Experiments

In this section, we evaluate AMP on different types of clusters and models. Besides a homogeneous setup, we study two heterogeneous setups, one with a heterogeneous cluster and one with a heterogeneous model. We use the state-of-the-art Megatron heuristics as our baseline [19].

### 4.1   Baseline

Megatron [23] is a runtime engine that supports 3D parallelism training on DNN models. It suggests several heuristics on how to train models with 3D parallelism faster. We summarize these heuristics in the following procedure.

Mgetraon explores all possible micro-batch sizes as in AMP. For each micro-batch size, it forces $tmp$ to be smaller than the smallest number of devices in a node. Then it selects parallelism degrees with the smallest $tmp \times pp$. It solves the device placement problem by prioritizing tensor model parallelism workers in the same node. If there are additional available intra-node connections, it places data parallelism workers next. It solves the pipeline layer assignment problem by either balancing the number of parameters or the number of layers in each stage.

Megatron is proposed based on observations on a homogeneous cluster and model setup. The model is composed of simple stacks of identical layers. The cluster is equipped with the same type of devices with the same bandwidths. Thus, We hypothesize under such as homogeneous setup, it is (near) optimal. To validate this hypothesis, we conduct experiments on a homogeneous setup (§ 4.3). Moreover, we are interested in how Megatron heuristics perform when the homogeneous assumption does not hold. Concretely, we ask two questions. What is optimal when the cluster is equipped with

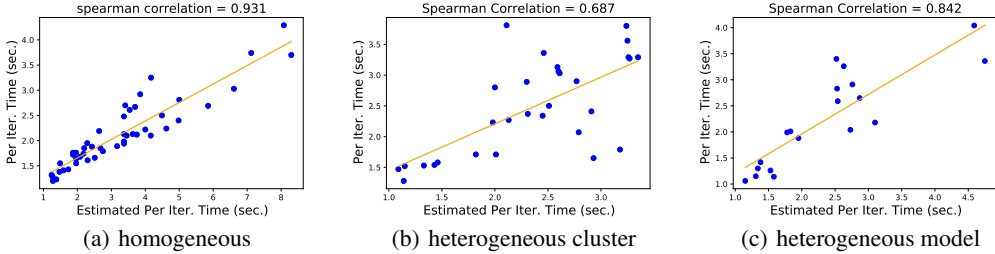

|  |  |  |
|:---:|:---:|:---:|
| (a) homogeneous | (b) heterogeneous cluster | (c) heterogeneous model |

Figure 2: Predicted strategy cost versus actual cost of top candidates. The Spearman correlation ranges from -1 to 1, and 0.5 is generally considered a reasonably strong positive correlation.

different types of devices and different bandwidths to each other? What is the effect of heterogeneity in models? To answer these, we design two sets of experiments under a heterogeneous cluster and a heterogeneous model in § 4.3.

## 4.2   Alternative Device Placement Algorithm

As described in Section 3.6, we adopt the heuristic in Megatron to solve the device placement algorithm in our main method. During the development time, we also explore an alternative device placement optimization method. Concretely, we adopt a Domino-tiling style device placement representation. Since this representation introduces much more possible strategies, enumerating them is costly. We instead use a Simulated Annealing(SA) algorithm to optimize the device placement aspect [25]. We name this method **SA**. Details can be found in § 6.3.

## 4.3   Experiment Setup

**Homogeneous.**   We conduct experiments using GPT-2 ($L = 24, H = 1024$) [20] on 4 AWS EC2 g4dn.12xlarge nodes with a global batch size 32. Each instance is equipped with 4 T4 GPUs with 50 Gbps PCIe connection intra-node bandwidth, and 50 Gbps inter-node bandwidth. In all experiments, we place instances in the same AWS placement group to make sure they have the full bandwidth.

**Heterogeneous cluster.**   To simulate heterogeneity in **both** different types of GPUs and different bandwidths, we ensemble a cluster using 3 AWS EC2 p3.8xlarge instances (V100 GPUs) and 1 g4dn.12xlarge (T4 GPUs) instance. Each p3.8xlarge instance is equipped with 4 V100 GPUs with approximately 170Gbps NVLink intra-node connection and 10 Gbps inter-node bandwidth. We evaluate the performance on GPT-2 ($L = 24, H = 1024$) with a global batch size 32.

**Heterogeneous model.**   We use a modified version of Transgan generator [12], which stacks transformer layers in the way that later layers have smaller hidden sizes. Specifically, we use 12 layers with dimensions 1024, and 12 layers with dimensions 16. We use 4 p3.8xlarge instances and a global batch size 64.

## 4.4   Evaluation Metric

We evaluate strategies by the running time per iteration. Specifically, we run each strategy by 40 iterations and take the average iteration time for the latest 20 iterations [27]. The underlying system is Deepspeed (built on top of the Megatron engine) [21] with fp16 optimization enabled.

## 4.5   Results and Discussion

**Homogeneous setup.**   In this experiment, we compare three methods: (1) *Megatron*: Megatron Heuristics, (2) *AMP*: our method described in § 3.6, and (3) *SA*: optimize the device placement aspect using the Simulated Annealing. Under homogeneous setup in § 4.3, we hypothesize that Megatron heuristics is (near) optimal. We also expect that our method can find such optimal strategies.

In our experiment setting, Megatron proposes 10 feasible strategies. To understand the difference in strategies between our proposal and the Megatron proposal, we evaluate the Top 10 strategies from

|  | Megatron-LM | RS | AMP | SA |
|---|---|---|---|---|
| Min (sec.) | 1.97 | 1.71 | **1.28** | 1.46 |
| Mean (sec.) | $2.62_{\pm 0.37}$ | $3.04_{\pm 0.82}$ | $\mathbf{1.73_{\pm 0.43}}$ | $2.29_{\pm 0.68}$ |

Table 4: Best and average strategies under heterogeneous cluster setup (scale: seconds).

|  | Megatron-LM | ours w/ parameter | ours w/ uniform | AMP |
|---|---|---|---|---|
| Min. (sec.) | 1.89 | 2.19 | 1.25 | **1.07** |
| Mean (sec.) | $1.94_{\pm 0.05}$ | $2.19_{\pm 0.0}$ | $1.25_{\pm 0.0}$ | $\mathbf{1.66_{\pm 0.67}}$ |

Table 5: Best and average strategy under heterogeneous model setting (scale: seconds)

our method. The results are presented in Table 3. We find that Megatron is indeed near-optimal and AMP can find such strategies (in fact slightly better). We analyze the selection procedure below.

Since the model can be fit into a single GPU, Megatron selects an extreme configuration - $dp = 16$. This may be sub-optimal because the communication in the data parallelism group is high. Suppose the micro-batch size is small, so that the bubble overhead in the pipeline parallelism may be hidden, it may be better to assign some workers to the pipeline parallelism. In fact, AMP finds that $dp = 4, pp = 4$ with the smallest micro-batch size 1 is a better choice. We also find that exploring more device placement strategies with SA finds worse results. This is because considering more device placement has limited influence in the homogeneous setup. For instance, whether the tensor model parallelism communication happens within a node or cross node shall not change the performance. However, it introduces much more strategies that make the optimization hard.

**Heterogeneity in the cluster.** In the heterogeneous cluster setup, devices have different connectivity to each other, and different execution speeds. We hypothesize that some heuristics from Megatron can lead to bad performance, but our method should be able to automatically address the heterogeneity in (especially) bandwidth since we explicitly model it (3.4). We are also interested in how much heterogeneity can influence the performance of strategies. Concretely, we examine how much Megatron and AMP improve compared to random strategies in the space. In this experiment, we have four methods to compare: Megatron, AMP, SA, and

- *RS*: Randomly selecting strategies without heuristics and a cost model.

We run RS for 50 iterations. Results are presented in Table 4. We find that the mean of random strategies is $1.16\times$ worse than that of Megatron, and $1.76\times$ worse than that of ours. By using our optimization procedure, we find strategies that achieve $1.54\times$ speedup compared to Megatron. Specifically, the heuristic of Megatron that minimize $mp \times pp$ is **sub-optimal**. When using micro-batch size 1, the best strategy shall be $dp = 2, pp = 8$, which results in a 1.28 second/iteration performance. However, since the model can be fit in a single GPU, Megatron selects dp=16, which results in 2.72 second/iteration performance. This strategy is much slower because some GPUs have a low bandwidth (10Gbps) to others. On the contrary, by modeling each GPU's bandwidth in Equation 3, our cost model penalizes this strategy heavily. We let the cost model to estimate $f(\cdot)$ for all strategies and find that AMP ranks this strategy in the $21^{st}$ place.

**Heterogeneity in the model.** When the layers in the model have different execution costs, the pipeline assignment methods in Megatron, which balances the number of layers or parameters, can not balance the actual workload. In such a setup, we expect our pipeline assignment can outperform these two. In this experiment, We compare four methods: (1) *Megatron*: Megatron with the uniform layer assignment, (2) *AMP*, (3) *ours w/ uniform*: the best strategy found by AMP but replaced with the uniform layer assignment, and (4) *ours w/ parameter*: the best strategy found by AMP but replaced with the uniform parameter assignment.

Results are presented in Table 5. AMP finds a $1.77\times$ faster strategy than Megatron ones. We analyze the speedup in two dimensions. First, if we only consider the micro-batch size and the parallelism

degrees aspect, the strategy found by AMP is $1.51\times$ faster than the one by Megatron. The reason is similar to the one in the heterogeneous cluster setting. Second, if we additionally consider the pipeline layer assignment aspect, the strategy gains an additional $1.17\times$ speedup.

To study the relationship between heterogeneity in the model and the system performance, we fix the parallelism degrees and vary only the pipeline layer assignment. Results are presented in 3. Since we only use transformer layers with two different hidden sizes, we simply define heterogeneity as:

$$heterogeneity = \frac{\text{h=16 layers time}}{\text{h=1024 layers time}} \tag{9}$$

**Cost model accuracy**    To correctly select the top strategies, AMP must rely on an accurate cost model. We consider the cost model to be accurate if it can correctly rank different strategies, where the absolute value of its prediction is of less importance. To study the accuracy, we obtain more ground truth $f(\cdot)$ (more than 10 for the main result) and compare them with the estimated $f(\cdot)$ by the cost model. The results are presented in Figure 2. We find that generally a lower estimate $f(\cdot)$ corresponds to a lower actual $f(\cdot)$. More importantly, strategies with low actual $f(\cdot)$ are ranked among the top. Concretely, the best strategy is ranked $3^{rd}$, $2^{nd}$, and $1^{st}$ in the three settings. As a result, the best strategy can be selected because AMP runs several real trials at the end of the optimization procedure (Algorithm 1). While AMP outperforms the baseline as it needs to launch 10 real trials, we would like to improve our cost model in the future so that fewer real trials are needed (each trial takes around 1-2 minutes in our setting).

**Optimization cost.**    In the heterogeneous model experiment, AMP takes 140.7 seconds for profiling and 235.6 seconds for 65 iterations of the optimization procedure. In particular, 230.8 seconds are spent in the dynamic programming algorithm. We also launch several real trials for top predicted candidates to select the final best one. In total, the cost of our optimization is around 5-10 minutes with several real trials (around $1-2$ minutes per trial).

**Limitations.**    AMP does not model memory footprint. Consequently, it can suggest strategies that do not fit in the device's memory, and launch unnecessary real trials. In our experiments, a small number of top strategies are out of memory (*i.e.,* $7^{th}$ and $9^{th}$ of the top 10 strategies in the TransGAN experiment). Additionally, the solution for the pipeline assignment problem is not efficient enough.

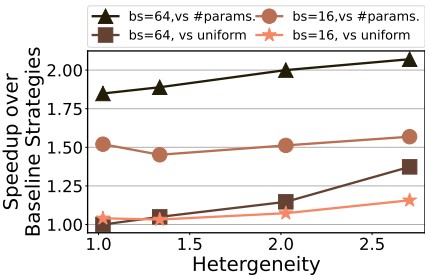

Figure 3: Speedup against heterogeneity versus two Megatron layer assignment methods. Numbers obtained when a top parallelism strategy: mp=1, pp=4, dp=4.

In our experiments, it takes the major time of the whole optimization procedure. When the number of layers $L$ goes up, the running time is increasing in the $\mathcal{O}(L^4)$. In the future, we aim to either improve the dynamic programming solution itself, or combine the solution with techniques such as operator clustering to keep $L$ low [2, 29].

## 5   Conclusion

We present AMP, an automatic algorithm that can find model parallel training strategies with high system throughput. AMP is equipped with an expert-design cost model that considers cluster and model configurations. We show that when heterogeneity exists in cluster or model setup, current heuristics are sub-optimal. Our automatic algorithm can find strategies with $1.51\times$ and $1.76\times$ speedup within similar budgets.

## Acknowledgements

We would like to thank Sangkeun Choe and Aurick Qiao for their comments in the early stage of the paper. We thank Tailin Zhang and Tianyi Yu for their help in deriving the pipeline objective and the dynamic programming solution. We thank all reviewers for their invaluable comments and feedback. This research was supported by NSF IIS1563887, NSF CCF1629559, NSF IIS1617583, NGA HM04762010002, NSF IIS1955532, NSF CNS2008248, NSF IIS2123952, and NSF BCS2040381.

