# References

[1] Jean-Baptiste Alayrac, Jeff Donahue, Pauline Luc, Antoine Miech, Iain Barr, Yana Hasson, Karel Lenc, Arthur Mensch, Katie Millican, Malcolm Reynolds, et al. Flamingo: a visual language model for few-shot learning. *arXiv preprint arXiv:2204.14198*, 2022.

[2] Kevin Aydin, MohammadHossein Bateni, and Vahab Mirrokni. Distributed balanced partitioning via linear embedding. In *Proceedings of the Ninth ACM International Conference on Web Search and Data Mining*, pages 387–396, 2016.

[3] Dimitris Bertsimas and John Tsitsiklis. Simulated annealing. *Statistical science*, 8(1):10–15, 1993.

[4] Tom B Brown, Benjamin Mann, Nick Ryder, Melanie Subbiah, Jared Kaplan, Prafulla Dhariwal, Arvind Neelakantan, Pranav Shyam, Girish Sastry, Amanda Askell, et al. Language models are few-shot learners. In *Advances in Neural Information Processing Systems*, volume 33, pages 1877–1901. Curran Associates, Inc., 2020.

[5] Henggang Cui, Hao Zhang, Gregory R Ganger, Phillip B Gibbons, and Eric P Xing. Geeps: Scalable deep learning on distributed gpus with a gpu-specialized parameter server. In *Proceedings of the eleventh european conference on computer systems*, pages 1–16, 2016.

[6] Jacob Devlin, Ming-Wei Chang, Kenton Lee, and Kristina Toutanova. Bert: Pre-training of deep bidirectional transformers for language understanding. *arXiv preprint arXiv:1810.04805*, 2018.

[7] Shiqing Fan, Yi Rong, Chen Meng, Zongyan Cao, Siyu Wang, Zhen Zheng, Chuan Wu, Guoping Long, Jun Yang, Lixue Xia, et al. Dapple: A pipelined data parallel approach for training large models. In *Proceedings of the 26th ACM SIGPLAN Symposium on Principles and Practice of Parallel Programming*, pages 431–445, 2021.

[8] Kaiming He, Xiangyu Zhang, Shaoqing Ren, and Jian Sun. Deep residual learning for image recognition. In *Proceedings of the IEEE conference on computer vision and pattern recognition*, pages 770–778, 2016.

[9] Qirong Ho, James Cipar, Henggang Cui, Seunghak Lee, Jin Kyu Kim, Phillip B Gibbons, Garth A Gibson, Greg Ganger, and Eric P Xing. More effective distributed ml via a stale synchronous parallel parameter server. In *Advances in neural information processing systems*, pages 1223–1231, 2013.

[10] Yanping Huang, Youlong Cheng, Ankur Bapna, Orhan Firat, Dehao Chen, Mia Chen, HyoukJoong Lee, Jiquan Ngiam, Quoc V Le, Yonghui Wu, et al. Gpipe: Efficient training of giant neural networks using pipeline parallelism. *Advances in neural information processing systems*, 32:103–112, 2019.

[11] Zhihao Jia, Matei Zaharia, and Alex Aiken. Beyond data and model parallelism for deep neural networks. *SysML 2019*, 2019.

[12] Yifan Jiang, Shiyu Chang, and Zhangyang Wang. Transgan: Two pure transformers can make one strong gan, and that can scale up. *Advances in Neural Information Processing Systems*, 34, 2021.

[13] Yimin Jiang, Yibo Zhu, Chang Lan, Bairen Yi, Yong Cui, and Chuanxiong Guo. A unified architecture for accelerating distributed {DNN} training in heterogeneous gpu/cpu clusters. In *14th {USENIX} Symposium on Operating Systems Design and Implementation ({OSDI} 20)*, pages 463–479, 2020.

[14] Dmitry Lepikhin, HyoukJoong Lee, Yuanzhong Xu, Dehao Chen, Orhan Firat, Yanping Huang, Maxim Krikun, Noam Shazeer, and Zhifeng Chen. Gshard: Scaling giant models with conditional computation and automatic sharding. *arXiv preprint arXiv:2006.16668*, 2020.

[15] Mu Li, David G Andersen, Jun Woo Park, Alexander J Smola, Amr Ahmed, Vanja Josifovski, James Long, Eugene J Shekita, and Bor-Yiing Su. Scaling distributed machine learning with the parameter server. In *11th {USENIX} Symposium on Operating Systems Design and Implementation ({OSDI} 14)*, pages 583–598, 2014.

[16] Zhuohan Li, Siyuan Zhuang, Shiyuan Guo, Danyang Zhuo, Hao Zhang, Dawn Song, and Ion Stoica. Terapipe: Token-level pipeline parallelism for training large-scale language models. *arXiv preprint arXiv:2102.07988*, 2021.

[17] Azalia Mirhoseini, Hieu Pham, Quoc V Le, Benoit Steiner, Rasmus Larsen, Yuefeng Zhou, Naveen Kumar, Mohammad Norouzi, Samy Bengio, and Jeff Dean. Device placement optimization with reinforcement learning. In *International Conference on Machine Learning*, pages 2430–2439. PMLR, 2017.

[18] Deepak Narayanan, Aaron Harlap, Amar Phanishayee, Vivek Seshadri, Nikhil R Devanur, Gregory R Ganger, Phillip B Gibbons, and Matei Zaharia. Pipedream: generalized pipeline parallelism for dnn training. In *Proceedings of the 27th ACM Symposium on Operating Systems Principles*, pages 1–15, 2019.

[19] Deepak Narayanan, Mohammad Shoeybi, Jared Casper, Patrick LeGresley, Mostofa Patwary, Vijay Korthikanti, Dmitri Vainbrand, Prethvi Kashinkunti, Julie Bernauer, Bryan Catanzaro, et al. Efficient large-scale language model training on gpu clusters using megatron-lm. In *Proceedings of the International Conference for High Performance Computing, Networking, Storage and Analysis*, pages 1–15, 2021.

[20] Alec Radford, Jeffrey Wu, Rewon Child, David Luan, Dario Amodei, Ilya Sutskever, et al. Language models are unsupervised multitask learners. *OpenAI blog*, 1(8):9, 2019.

[21] Jeff Rasley, Samyam Rajbhandari, Olatunji Ruwase, and Yuxiong He. Deepspeed: System optimizations enable training deep learning models with over 100 billion parameters. In *Proceedings of the 26th ACM SIGKDD International Conference on Knowledge Discovery & Data Mining*, pages 3505–3506, 2020.

[22] Alexander Sergeev and Mike Del Balso. Horovod: fast and easy distributed deep learning in tensorflow. *arXiv preprint arXiv:1802.05799*, 2018.

[23] Mohammad Shoeybi, Mostofa Patwary, Raul Puri, Patrick LeGresley, Jared Casper, and Bryan Catanzaro. Megatron-lm: Training multi-billion parameter language models using model parallelism. *arXiv preprint arXiv:1909.08053*, 2019.

[24] Jakub M Tarnawski, Deepak Narayanan, and Amar Phanishayee. Piper: Multidimensional planner for dnn parallelization. *Advances in Neural Information Processing Systems*, 34, 2021.

[25] Peter JM Van Laarhoven and Emile HL Aarts. Simulated annealing. In *Simulated annealing: Theory and applications*, pages 7–15. Springer, 1987.

[26] Samuel Williams, Andrew Waterman, and David Patterson. Roofline: an insightful visual performance model for multicore architectures. *Communications of the ACM*, 52(4):65–76, 2009.

[27] Hao Zhang, Peng Wu, Zhijie Deng, Christy Li, Qirong Ho, Aurick Qiao, Zeya Wang, and Eric P Xing. Autodist: Acomposable and automated synchronization system for distributed deep learning.

[28] Hao Zhang, Zeyu Zheng, Shizhen Xu, Wei Dai, Qirong Ho, Xiaodan Liang, Zhiting Hu, Jinliang Wei, Pengtao Xie, and Eric P Xing. Poseidon: An efficient communication architecture for distributed deep learning on {GPU} clusters. In *2017 USENIX Annual Technical Conference (USENIX ATC 17)*, pages 181–193, 2017.

[29] Lianmin Zheng, Zhuohan Li, Hao Zhang, Yonghao Zhuang, Zhifeng Chen, Yanping Huang, Yida Wang, Yuanzhong Xu, Danyang Zhuo, Joseph E Gonzalez, et al. Alpa: Automating inter- and intra-operator parallelism for distributed deep learning. *arXiv preprint arXiv:2201.12023*, 2022.


Figure 4: Communication and computation overlap illustration using 2 workers. The first stage sends the activation to the second stage once it finishes the first micro-batch with time $t_1$, while continuing the next micro-batch computation.

# 6 Supplementary Materials

## 6.1 Computation and communication overlap

We consider overlapping in the P2P communication as: the sender sends the message to the receiver, while continuing its computation. The receiver needs to wait for the message before continuing its computation. This overlapping is illustrated in Figure 4.

## 6.2 Optimization procedure details

We present the rest details of Algorithm 1. We briefly describe how each sub-procedure is implemented here:

1. *placement()* generates the device placement using the heuristic in Megatron.
2. *enumerate_degrees()* takes in the cluster $\mathcal{C}$ information, and outputs all possible parallelism degrees, each with format $(pp, dp, tmp)$ with constraints that $pp \times dp \times tmp = |\mathcal{D}|$.
3. *pipe_ast()* takes in the model $\mathcal{W}$ information, the number of stages $d.pp$. It generates the per layer cost, and per edge cost using formulas in section 3.4, and uses the dynamic programming algorithm in section 3.5 to solve for an optimal layer assignment.
4. *estimate()* takes in the optimal layer assignment, and generates the final estimate time using Equation 4.

## 6.3 Randomized Optimization

In the layer assignment problem, we leverage the structure that only continuous layers can be assigned to the same stage, which enables a solution with optimality and polynomial runtime. However, other aspects exhibit less structure that allows us to leverage a deterministic algorithm. Moreover, these dimensions compose a large space that prohibits us from simply enumerating all possibilities. Specifically, there are exponentially many possible device placements [17]. At a high level, we would like to optimize a cost function over a discrete domain with a large support. One effective family of algorithms to solve this problem is known as *Simulated Annealing* (SA) [3], where it explores states (strategies in our semantics) in a neighbor-to-neighbor fashion and gradually improves the quality. A neighbor of a state is produced by conservatively changing the current state.

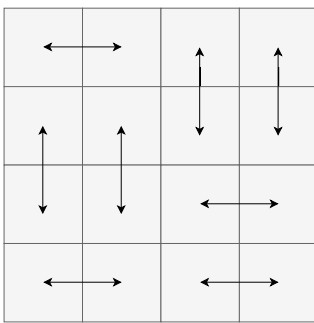

Figure 5: A possible device placement when using a $4 \times 4$ device grid, and 8 pipeline stages. Each device is represented as a square. Squares connected with arrows are associated with the same pipeline stage (a domino).

**Algorithm 2:** randomized optimization procedure

**Input:** iteration, budget

1  t = 1.0 // Temperature parameter
2  s = initialize_strategy()
3  record = set()
4  cost = estimate(s) // cost model
5  **for** *i in iteration* **do**
6      t = cool_down(t, i)
7      next = s.copy()
8      **if** *randn() > 0.5* **then**
9          next.tmp = sample_mp(s.dp) // vary tmp
10     **else**
11         next.dp = sample_dp(s.tmp) // vary dp
12     next.mbs = sample_mbs(next.dp)
13     next.placement = sample_domino(next.dp, next.tmp)
14     next.a = pipe_ast(s) // pipeline assignment
15     next_cost = estimate(next)
16     acc_prob = $e^{\frac{min(cost-next\_cost,0)}{t}}$
17     **if** *randn() < acc_prob* **then**
18         s = next // accept
19         record.add(s)
20         cost = next_cost

   /* run top predicted strategies                                 */
21 best_s = run(record, budget)
22 **return** best_s

---

Specifically, we use a SA algorithm that varies only $dp$ or $tmp$ at a time (i.e. strategies with the same $dp$ or $tmp$ are neighbors), guided by the cost model in section 3.4 to iteratively find a good strategy. To optimize the device placement aspect given $tmp$ and $dp$, we consider a problem setup similar to the *Domino Tiling* problem: we place devices as a 2d mesh and tile it using dominos of size $tmp \times dp$ either horizontally or vertically. Each domino represents ranks with the same pipeline stage. [4]. An example domino tiling scheme is shown in Figure 5. The described optimization procedure is presented at algorithm 2.

### 6.4 Proof of the dynamic programming solution

At a cut with position i', equation 6 can be rewritten as

$$g(i') = (gas - 1) \cdot \max\{0, \max\{t_{1,i'}, t_{2,i'}\} - m\} + \sum_{i=1}^{pp-1} e_i + \sum_{i=1}^{pp} t_i \tag{10}$$

The associate cost stored at $dp[i'][j-1][max\{t_{2,i'}, m\}]$ is:

$$c_{i'} = (gas - 1) \cdot \max\{0, t_{1,i'} - max\{t_{2,i'}, m\}\} + \sum_{i=1}^{pp-2} e_i + \sum_{i=1}^{pp-1} t_i \tag{11}$$

Claim:
$$max\{0, t_{1,i'} - max\{t_{2,i'}, m\}\} + max\{0, t_{2,i'} - m\} \tag{12}$$

$$= max\{0, max\{t_{1,i'}, t_{2,i'}\} - m\} \tag{13}$$

Prove claim by enumerate all possibility:

---

[4] Observe that communication within a domino is DP or MP, which is both more intense than PP communication. We place devices with higher bandwidth closer in the 2d mesh to reduce the communication time. We further assume that vertically connected devices in a domino form a MP group, whereas horizontal ones form a DP group. Thus, each tiling exactly corresponds to a device placement function.

1. $t_{1,i'} < t_{2,i'} < m$: Equation 12 = 0 + 0 = 0, Equation 13 = 0
2. $t_{1,i'} < m < t_{2,i'}$: Equation 12 = 0 + $t_{2,i'}$ - m, Equation 13 = $t_{2,i'}$ - m
3. $t_{2,i'} < t_{1,i'} < m$: Equation 12 = 0 + 0, Equation 13 = 0
4. $t_{2,i'} < m < t_{1,i'}$: Equation 12 = $t_{1,i'}$ - m + 0, Equation 13 = $t_{1,i'}$ - m.
5. $m < t_{1,i'} < t_{2,i'}$: Equation 12 = 0 + $t_{2,i'}$ - m, Equation 13=$t_{2,i'}$ - m.
6. $m < t_{2,i'} < t_{1,i'}$: Equation 12 = $t_{1,i'} - t_{2,i'} + t_{2,i'}$ - m = $t_{1,i'}$ - m, Equation 13 = $t_{1,i'}$ - m.

Using the claim:

$$g(i') = c_{i'} + (gas - 1) * max\{0, t_{2,i'} - m\} + t_{2,i'} + e_{i',j} \tag{14}$$

## 6.5 Experiment Details

In this section, we provide detailed experiment setup and results to help understand the performance of each training strategy, and how different methods find different top strategies. In particular, we provide the Top 5 strategies proposed by Megatron and AMP.

### 6.5.1 Homogeneous setting

**Model architecture** We use a 24 layers GPT-2 model, where layer 3-26 are transformer layers, and the rest are lambda functions or embedding layers. We use hyper-parameters: hidden size 1024, sequence length 1024, vocabulary size 52256, and batch size 32.

| mbs | pipeline layer assignment | tmp | pp | run time |
|-----|---------------------------|-----|-----|----------|
| 1 | [0, 30] | 1 | 1 | **1.32** |
| 2 | [0, 14, 30] | 1 | 2 | 1.37 |
| 2 | [0, 30] | 2 | 1 | 1.58 |
| 4 | [0, 14, 30] | 1 | 2 | 1.63 |
| 4 | [0, 30] | 2 | 1 | 1.53 |

Table 6: Top 5 candidates by Megatron under homogeneous setup (scale: seconds).

| mbs | pipeline layer assignment | tmp | pp | run time |
|-----|---------------------------|-----|-----|----------|
| 1 | [0, 15, 30] | 1 | 2 | 1.28 |
| 1 | [0, 9, 15, 21, 30] | 1 | 4 | **1.20** |
| 1 | [0, 6, 9, 12, 15, 18, 21, 24, 30] | 1 | 8 | 1.23 |
| 2 | [0, 15, 30] | 1 | 2 | 1.38 |
| 1 | [0, 4, 6, 8, 10, 12, 14, 16, 17, 18, 19, 20, 21, 22, 23, 25, 30] | 1 | 16 | 1.55 |

Table 7: Top 5 candidates by AMP under homogeneous setup (scale: seconds).

### 6.5.2 Heterogeneous Cluster

**Model architecture** We use the same model configuration as in the homogeneous setup.

| mbs | pipeline layer assignment | tmp | pp | run time |
|-----|---------------------------|-----|-----|----------|
| 2 | [0, 14, 30] | 1 | 2 | 2.27 |
| 4 | [0, 14, 30] | 1 | 2 | 2.32 |
| 8 | [0, 8, 14, 20, 30] | 1 | 4 | **1.97** |
| 8 | [0, 14, 30] | 2 | 2 | 2.43 |
| 16 | [0, 8, 14, 20, 30] | 2 | 4 | 2.34 |

Table 8: Top 5 candidates by Megatron under heterogeneous cluster setup (scale: seconds).

| mbs | pipeline layer assignment | tmp | pp | run time |
|-----|---------------------------|-----|----|----------|
| 1 | [0, 7, 14, 20, 30] | 1 | 4 | 1.47 |
| 1 | [0, 5, 9, 12, 15, 18, 21, 24, 30] | 1 | 8 | **1.28** |
| 1 | [0, 3, 5, 7, 9, 11, 13, 15, 16, 17, 19, 20, 21, 22, 23, 25, 30] | 1 | 16 | 1.52 |
| 2 | [0, 8, 14, 20, 30] | 1 | 4 | 1.52 |
| 2 | [0, 5, 9, 12, 14, 17, 20, 23, 30] | 1 | 8 | 1.54 |

Table 9: Top 5 candidates by AMP under heterogeneous cluster setup (scale: seconds).

### 6.5.3 Heterogeneous Model

**Model architecture**  We use a TransGAN Generator with 24 transformer layers, where layer 4-15, 42-53 are transformer layers, and the rest are lambda functions or embedding layers. We use hyper-parameters: hidden size 1024, bottom width 9, batch size 64, 12 transformer layers for stage 1, and 12 transformer layers for stage 6.

| mbs | pipeline layer assignment | tmp | pp | run time |
|-----|---------------------------|-----|----|----------|
| 1 | [0, 41, 56] | 1 | 1 | **1.89** |
| 2 | [0, 9, 41, 47, 56] | 1 | 2 | 1.99 |

Table 10: Top candidates by Megatron under heterogeneous model setup (scale: seconds). Strategies provided by Megatron are out of memory for larger micro batch size due to its pipeline layer assignment method.

| mbs | pipeline layer assignment | tmp | pp | run time |
|-----|---------------------------|-----|----|----------|
| 1 | [0, 12, 44, 48, 56] | 1 | 4 | **1.07** |
| 2 | [0, 12, 44, 48, 56] | 1 | 4 | 1.08 |
| 2 | [0, 5, 8, 11, 14, 42, 43, 44, 45, 46, 47, 48, 49, 50, 51, 52, 56] | 1 | 16 | 1.13 |
| 1 | [0, 5, 8, 11, 14, 42, 43, 44, 45, 46, 47, 48, 49, 50, 51, 52, 56] | 1 | 16 | 1.30 |
| 2 | [0, 7, 12, 42, 44, 47, 49, 51, 56] | 1 | 8 | 1.39 |

Table 11: Top 5 candidates by AMP under heterogeneous model setup (scale: seconds).