# OpenReview forum: "AMP: Automatically Finding Model Parallel Strategies with Heterogeneity Awareness"
_NeurIPS.cc/2022/Conference — NeurIPS 2022 Accept_

### Official Review · Reviewer_9vj5 · 2022-07-05

**Rating:** 5
**Confidence:** 2
**Soundness:** 3 good
**Presentation:** 2 fair
**Contribution:** 2 fair

**Summary:**

This paper presents a strategy in model-parallel setting, where the model is being trained in a cluster. The strategy considers heterogeneity of computational resources, diverse characteristics of different layers, communication costs in networks, and uses a dynamic programming algorithm to intelligently assign layer-stages.


**Questions:**

1) The reviewer is interested in more study on how accurate the cost model is, for example, what if we replace the analytical approximation with an oracle cost model?
2) Compared with previous work like Alpa which uses homogeneous setting, how far does the proposed method go besides adding more terms in the dynamic programming equaltion?

**Ethics Review Area:**

["I don’t know"]

**Limitations:**

As the author stated, memory footprint is right now not under consideration. However, the reviewer does not consider this as a significant drawback.

**Strengths And Weaknesses:**

Strength:
1) A dynamic programming-based algorithm is prposed to find out the optimial placement strategy that is significantly time-efficient that bruteforce exhausive search;
2) The paper proposes an analytical cost model that saves the cost of estimating the run time of each layer on actual hardware;
3) Experiments show that the strategy is highly effective and provides concrete speedup for end-to-end workloads.

Weakness:
1) The paper doesn't present comprehensive ablation study of the effectiveness of the cost model.
2) The paper seems to overlook some latest advances in distributed training and model parallelism, for example, Alpa [1] that considers homogeneous setting but still formulates the problem similarly, while having heterogeneity seems to be a plain extension. Would love to hear from the reviewer about this particular comparison.

[1] Zheng L, Li Z, Zhang H, Zhuang Y, Chen Z, Huang Y, Wang Y, Xu Y, Zhuo D, Gonzalez JE, Stoica I. Alpa: Automating Inter-and Intra-Operator Parallelism for Distributed Deep Learning. arXiv preprint arXiv:2201.12023. 2022 Jan 28.

---

> ### Author Response · Authors · 2022-08-02
> **Response to reviewer 9vj5**
>
> We thank the reviewer for the constructive feedback on the cost model accuracy and  a comparison to Alpa. We summarize the questions here and respond to them individually:
>
> $\textbf{Question 1}$: What is the accuracy of the cost model? What if we can obtain an oracle cost model?
>
> $\textbf{Reponse}$: We consider the cost model to be “accurate” if it can correctly rank different strategies. The spearman correlation of our cost model is 0.812, 0.719, and 0.867 in our three experiments, where 0.5 is usually considered a reasonably strong positive correlation. We also updated plots showing the prediction and ground truth per iteration time in the revised manuscript (Please see Figure 2 in the revised manuscript).
>
> An oracle cost model will further minimize the optimization cost of our procedure by guaranteeing the best predicted strategy is always the real best one. In our method, we evaluate several top-predicted candidates at the end of the optimization since our cost model is not 100% accurate. For instance, in the first experiment, the best candidate is ranked 5th by our cost model. While the baseline method needs around 10 real trials, we would like to further improve the accuracy of our cost model in the future.
>
> $\textbf{Question 2}$: Is AMP a plain extension to latest work built to homogeneous settings like Alpa[1]?
>
> $\textbf{Response}$: We would like to highlight that AMP’s cost model is not a simple extension of one from a homogeneous setting.  Going from homogeneous settings to heterogeneous settings often requires changes in the fundamental assumptions (e.g. regularity assumptions in the model or cluster). This often requires new designs to the entire workflow,  where adding several terms in the cost model alone is insufficient.
>
> For the concrete system - Alpa, we would like to clarify three points.
>
> (i) Alpa is a concurrent work of AMP.
>
> (ii) Alpa relies on two fundamental regularity assumptions. First, Alpa assumes that the logical device mesh is 2-dimensional. For a 4x4 cluster, Alpa’s device mesh only considers the possibility of 2x8, 1x16, 4x4, 8x2, or 16x1. It could not handle a heterogeneous mesh shape such as 1x4 (g4dn.12xlarge) + 3x4 nodes (p3.8xlarge), which AMP is tested on. Second, Alpa assumes a uniform bandwidth in the same device mesh dimension. On the contrary, AMP does not have such assumptions on the bandwidth matrix (detailed in the problem setup in Section 3.1) and can handle general device connectivity. For instance, in the 1x4 + 3x4 testbed, the inter-node connection is not uniform (10 Gbps for the 3x4, and 50 Gbps for the 1x4). These two fundamental assumptions would make it hard to extend Alpa to handle general heterogeneity.
>
> (iii) Alpa requires tuning the micro-batch sizes manually. This is important in heterogeneous environments since different GPUs can have very different throughput against different micro-batch sizes (e.g., the arithmetic intensity issue). It would need efforts to incorporate micro-batch size in the problem setup of Alpa.
>
>
> [1] Zheng L, Li Z, Zhang H, Zhuang Y, Chen Z, Huang Y, Wang Y, Xu Y, Zhuo D, Gonzalez JE, Stoica I. Alpa: Automating Inter-and Intra-Operator Parallelism for Distributed Deep Learning. arXiv preprint arXiv:2201.12023. 2022 Jan 28.

---

### Official Review · Reviewer_tyEZ · 2022-07-08

**Rating:** 6
**Confidence:** 3
**Soundness:** 4 excellent
**Presentation:** 3 good
**Contribution:** 3 good

**Summary:**

This paper presents a method to automatically find the optimal parallelization strategy for large model training. Unlike previous frameworks, the proposed model considered the heterogeneity of the model and hardware. The parallelization problem is defined as the parallelism degrees, micro batch size, device assignment, and the arrangement of the layers in the pipeline. The authors built a cost model to estimate the cost of the candidate strategy. The parallelism degrees and micro batch size are selected through enumeration. The device assignment is solved by applying a heuristic while the pipeline arrangement is found through dynamic programming.

**Questions:**

1. Could you provide extra information on the accuracy of the proposed cost model? For example, the correlation coefficient against the real latency. Could you also indicate how will the accuracy of the cost model affect the result?

2. Could you comment on whether it is possible to perform joint optimization and how does it compare with the discrete optimization of each subproblem?

**Limitations:**

The author discussed the limitation in terms of the lack of memory model.

**Strengths And Weaknesses:**

## Strength

1. This paper present a detailed cost model for parallelization strategy on heterogeneous models and hardware. The difference in hardware is considered as different bandwidths and the profiling result of the layer execution time.

2. The pipeline arrangement is modeled as a dynamic programming problem and given a determined solution.

3. The ablation study shows the effectiveness of each component optimized by the proposed method.


## Weakness

1. **The accuracy of the cost model.** The evaluation focuses on the iteration latency. But as a major contribution of this paper, it is also important to understand the accuracy of the cost model and how will it affect the strategies found by the method. Based on what is shown in Figure 2, it seems that the estimation for most strategies is optimistic, especially for the optimal one indicated by the estimation.

2. **Discrete optimization.** The authors decompose the strategies into four different subproblems and separately optimize for each problem. Will this lead to a suboptimal solution compared with a joint optimization method?

---

> ### Author Response · Authors · 2022-08-02
> **Reponse to Reviewer tyEZ**
>
> We thank the reviewer for the constructive feedback on the cost model validation and possible joint optimization. We summarize the questions here and respond to them individually:
>
> $\textbf{Question 1}$: What is the accuracy of the cost model, and how will this affect the result?
>
> $\textbf{Response}$: The spearman correlation is 0.812, 0.719, 0.867 in our three experiment setups. The Spearman correlation ranges from -1 to 1, and 0.5 is generally considered a reasonably strong positive correlation. We also included plots to help understand the accuracy of our cost model qualitatively (please see Figure 2 in the revised manuscript).
>
> We consider the cost model to be “accurate” if it can correctly rank different strategies, where the absolute value of its prediction is of less importance. Since we evaluate several top candidates at the end, an accurate cost model will need fewer real trials. For instance, in the first setup, the best strategy is ranked 5th by our cost model. While we also outperforms the baseline since it needs to launch 10 real trials, we would like to improve our cost model in the future so that fewer real trials are needed (each trial takes around 1-2 minutes in our setting).
>
> $\textbf{Question 2}$: Is it possible to perform joint optimization over the four aspects?
>
> $\textbf{Response}$: We would like to clarify that we are performing joint optimization. We have updated an optimization algorithm in the revised manuscript. The outer loop enumerates parallelism degrees and micro-batch size, while the inner loop solves the pipeline layer assignment problem conditioned on the current parallelism degrees and micro-batch size.

---

> > ### Comment · Reviewer_tyEZ · 2022-08-07
> > **Response checked**
> >
> > Thanks for your response! I found the updated plot for the accuracy model helpful for understanding.
> >
> > I think all my concerns are addressed by the authors' responses.

---

### Official Review · Reviewer_5Naa · 2022-07-11

**Rating:** 6
**Confidence:** 5
**Soundness:** 3 good
**Presentation:** 2 fair
**Contribution:** 3 good

**Summary:**

This paper proposes a method to automatically determine the parallel strategy that includes data parallelism (DP), pipeline parallelism (PP), and tensor model parallelism (TMP) over a distributed runtime with heterogeneous computation and communication resources.

**Questions:**

- In line 116, does the tensor refer to output activations from layer i?

- Would it be possible to be more concrete about the optimization procedure? For example, can the dynamic programming approach explore all the space? What is the complexity of the optimization?

- Would it be possible to provide the execution time of the optimization procedure in the experiments?

**Limitations:**

It is helpful to note the limitation on the lack of memory constraints in the cost model.

**Strengths And Weaknesses:**

Strengths:

- Automatically determining the combined parallel strategy is an interesting and important problem for distributed machine learning considering the advanced of giant foundation models.

- The formalization of the problem and the proposed cost model is reasonable.

Weaknesses:
- The presentation of the methodology part can be further polished; some of the details are difficult to follow for the first time. For example, a table defining all the variables demanded in the cost model will make Section 3 easier to follow; and there should be a clear discussion about how communication and computation are considered as overlapped execution; perhaps a Gantt chart will be helpful to illustrate this.

- The optimization procedure is not clearly stated in Section 3.6. In fact, Section 3.6 is just a sketch instead of a concrete discussion of the technique. Many details are missing for optimization.

- The experiments are conducted on a relatively small GPT-2 model with a small batch-size; the cluster scale is also small with 4X4 GPUs in total. In fact, the practical challenge for automatic layout is for very large clusters with hundreds or even thousands of GPUs. In the current scope of experimental results, it is difficult to verify the effectiveness of the proposed method.

---

> ### Author Response · Authors · 2022-08-02
> **Reponse to Reviewer 5Naa**
>
> We thank the reviewer for the constructive feedback on the presentation details and optimization procedure. We summarize the questions here and respond to them individually:
>
> $\textbf{Question 1}$: The presentation for Section 3 can be further polished.
>
> $\textbf{Reponse}$: We improve the presentation of AMP’s methodology, by adding (i) optimization algorithm details; (ii) a table of variable definitions; (iii) a Gantt chart (please see the revised manuscript). We currently place some materials in the appendix for the page limit and will arrange them for the main content.
>
> $\textbf{Question 2}$: The optimization algorithms should contain more details: (i) The space of the dynamic programming algorithm; (ii) The complexity of the overall optimization; (iii) The wall clock execution time for the optimization.
>
> $\textbf{Reponse}$:  We include a concrete optimization algorithm in the revised manuscript. Concretely: (i) The dynamic programming approach is only used to solve for optimal layer assignment, which is a subspace in our search space. (ii) the overall complexity is bounded by $\mathcal{O}(gbs^{\frac{1}{2}}|\mathcal{D}|^2L^4)$. ($gbs$: global batch size, $|\mathcal{D}|$: number of devices, $L$: number of layers) (iii) The wall clock execution time for our optimization procedure is around 5 minutes plus several real trials at the end (around 1-2 minutes per trial ) (Section 4.5).
>
> $\textbf{Question 3}$: Experiments are conducted on a relatively small scale (4x4 cluster).
>
> $\textbf{Reponse}$:  We agree that our experiment scale is lagging behind what would be considered industry-scale (as mentioned by the reviewer, i.e., hundreds or even thousands of GPUs). However, we would like to highlight that one major motivation for considering heterogeneous clusters is to democratize the powerfulness of the large models, e.g., GPT-3. In practice, when there are not enough advanced GPUs, one can add more computation resources via leveraging older generations of GPUs. And AMP aims at such an application scenario. Thus, having thousands of advanced GPUs is not necessarily our targeted application scenario.
>
> Due to access and budget constraints, we were unable to finish experiments beyond 4x4 GPUs at the time of the paper submission. We'll extend to experiments to 8x8 clusters when we have access to larger clusters. However, we don’t see any fundamental reason why our cost model will fail to find a viable parallelism strategy when scaling to a bigger scale. One valid concern though is the efficiency of our search algorithm. We explicitly specified the complexity of our searching algorithm, i.e., $\mathcal{O}(gbs^{\frac{1}{2}}|\mathcal{D}|^2L^4)$. This term seems to suffer the efficiency issue when scaling to deeper models (with larger $L$) or clusters with more GPUs (with larger $D$).
> However, for deep models (with a larger $L$), we can adopt various operator clustering techniques used in [1-2] to keep it constant.
> Despite applications in extremely large clusters (with 1000s of GPUs), model parallelism on medium-scale clusters is enormous and also of great importance, and can straightforwardly benefit from AMP.
>
> $\textbf{Question 4}$: In line 116, does the tensor refer to output activations from layer $i$?
>
> $\textbf{Reponse}$: Yes, the tensor is the output activation from layer $i$. We have updated (please see the revised manuscript) the text to avoid confusion.
>
> [1] https://arxiv.org/pdf/1512.02727.pdf
> [2] https://www.usenix.org/conference/osdi22/presentation/zheng-lianmin

---

> > ### Comment · Reviewer_5Naa · 2022-08-08
> > **Thanks for the response!**
> >
> > I want to thank the author for their detailed response!
> >
> > I think the modification made the presentation of the paper much better to follow. I do appreciate the great effort the author has made!
> >
> > Meanwhile, I believe my concern about cluster scale could be addressed better:
> >
> > - If you could run experiments on cloud platform, could you rent more machine with less powerful GPU with the same budget of the current type of GPU?
> >
> > - Could you conduct some study about the relationship between the optimization time and cluster scale (to justify the theoretical claim of the complexity)?

---

> > > ### Author Response · Authors · 2022-08-09
> > > **Response to Reviewer 5Naa's additional comments**
> > >
> > > Response: We thank the reviewer for further constructive comments! To address the cluster scale concern, we further tested AMP's running time on TransGAN (a 24-layer Transformer model) over clusters with sizes of 2x2, 4x4, 8x8, and 16x16, i.e., up to 256 GPUs. The results are shown in the Table below:
> > >
> > > | Cluster size      | 2x2 | 4x4 | 8x8 | 16x16 |
> > > | ----------- | ----------- |----------- |----------- |----------- |
> > > |AMP solving time (in sec.)      | 57.98       |325.64| 890.3| 1627.4|
> > >
> > > Where we observe that AMP's solving time approximately grows linearly with the number of GPUs. Though our complexity analysis indicates the running time grows quadratically with the number of GPUs, the bound is not tight. And this explains why the actual solving time of AMP scales linearly with the number of GPUs.
> > >
> > > Even on a 16x16 cluster with 256 GPUs, AMP only takes 0.45 hours for finding the desirable parallelism strategy. Compared to the overall TransGAN training cost, which can take a few days, 0.45 hours is almost negligible. We thus believe that AMP enjoys good efficiency even when scaling over large clusters.

---

### Official Review · Reviewer_Nnic · 2022-07-12

**Rating:** 6
**Confidence:** 4
**Soundness:** 2 fair
**Presentation:** 2 fair
**Contribution:** 2 fair

**Summary:**

This paper provides a model to optimize large DL models. It takes into account TMP, DMP, and PMP. Moreover, the proposed model accounts for heterogeneity that exists across devices.

**Questions:**

Please address the weaknesses mentioned above.

**Limitations:**

* Major heterogeneity that exists today in clusters are in terms of GPU types, e.g., use of A100, V100 etc. Paper falls short of its claims

* It is unclear as to why only select few device characteristics such as bandwidth was used.

* Overall it is bunch of heuristics rather than a global and generic optimization framework. Perhaps the paper is more suited for a system’s conference such as OSDI/Eurosys/ATC

**Strengths And Weaknesses:**

Strengths
—-

* models device heterogeneity
* performs better than state of the art


Weakness
—-

* model is not generic. Lot of heuristics that have been integrated; thereby, exploring few possible optimizations
* For example, why was latency differences among devices not modelled. It is well known that both bandwidth and latency play a role
* Similarly, why not consider different generations of GPUs. Is not that more realistic?
* Does not really account for virtualisation and multi-tenent environment.
* moreover, the pruning strategies are not really clear. Overall it seems to me that most of the optimization can be specified by a rule-based system

---

> ### Author Response · Authors · 2022-08-02
> **Reponse to Reviewer Nnic**
>
> We thank the reviewer for the constructive and insightful feedback on the cost model and pruning strategy. We summarize the weakness and questions here and respond to them individually:
>
> $\textbf{Question 1}$: Cost model is not generic because it does not model: (i) latency; (ii) different generations of GPUs; (iii) virtualization, (iv) multi-tenant environments. The model may be specified by a rule-based system.
>
> $\textbf{Response}$: To the best of our knowledge, designing a 100% omnipotent cost model that models all the above aspects is a very hard problem. AMP takes a first step toward trading off the complexity of the cost model, searching efficiency, and the throughput of the found parallel strategy from a reasonably large search space. We explain how AMP handles (i)-(iv) below.
>
> (i) In the current cost model, we only model the bandwidth cost, which follows the BytePS system [1]. Based on our tests, only modeling bandwidth costs is viable for the cluster scale we experiment with, i.e., up to 4 nodes (4 GPUs on each node). We agree that when it comes to a larger scale, e.g., 100 nodes, latency costs start to become non-negligible. We will add a latency term in the later versions of our cost model as future work.
>
> (ii) We would like to clarify that AMP does consider different generations of GPUs. The per-layer execution cost is obtained by a few profiling trials with different Tensor Model Parallelism (TMP) degrees (section 3.4). By evaluating using the real hardware, we can handle the different speeds between generations of GPUs. We also model different bandwidths between different generations of GPUs by taking in the cluster bandwidth matrix as an argument for the cost model (section 3.1, 3.4). In the experiment for heterogeneous clusters, our model is tested with different generations of GPUs:  p3.8xlarge instances are equipped with Nvidia V100 GPUs and g4dn.12xlarge instances are equipped with Nvidia T4 GPUs. We have updated our text to clarify the GPU versions (Please see the revised manuscript).
>
> (iii) For virtualization, our current version of AMP runs on AWS, which indeed is on top of VMs.
>
> (iv) We agree that considering multi-tenant environment is important, but is out of the scope of this paper. We defer applying AMP in a multi-tenant environment as future work.
>
> As for “the most…can be ..by a rule-based system”, we respectfully disagree with the reviewer. We believe “rule-based systems” in this context roughly mean that they are $\textbf{static}$ to model and cluster setup.  In our experiment baseline, the set of Megatron Heuristics is a reasonably strong rule-based system, which uses uniform layer assignment strategies to deal with the layer assignment problem, but we show that it cannot handle more heterogeneous models. For instance, TransGAN has different workloads in different layers, and uniformly assigning layers is sub-optimal. Figure 4 shows that our method is more suitable than a rule-based system since it can $\textbf{automatically}$ find optimal strategies with respect to the model and cluster setup.
>
> $\textbf{Question 2}$: Pruning strategy is not clearly stated.
>
> $\textbf{Response}$: The “pruning” strategy refers to our dynamic programming algorithm. We have updated the text to avoid confusion.
>
> $\textbf{Question 3}$: The paper may be better suited for a system venue.
>
> $\textbf{Response}$: We would like to highlight the main contribution of AMP is its heterogeneity-aware cost model and its search algorithm, which is relevant to understanding and accelerating the computation of various ML models and user-specified tasks. Thus, we believe that NeurIPS is the right venue for AMP.
>
> [1] https://www.usenix.org/system/files/osdi20-jiang.pdf

---

### Author Response · Authors · 2022-08-02
**General Response**

We thank all reviewers for their positive feedback (scores: 6/6/6/5). We are encouraged that they all appreciated our paper for the following reasons: (i) our cost model of AMP captures the hardware and machine learning model heterogeneity; (ii) AMP determines the 3D parallelism strategy automatically and efficiently; (iii) the strategies returned by AMP lead to faster system throughput compared to the state-of-the-art systems, e.g., Megatron-LM. Each reviewer provided helpful suggestions to improve our manuscript that we address below along with additional experimental results and a revised paper manuscript (where the modified parts are highlighted in blue).

Based on the feedback, we mainly update the manuscript with (i) a cost model accuracy study; (ii) a table that summarizes the variables used in the cost model; (iii) a discussion on the overlap between computation and communication; (iv) a clearly written optimization procedure algorithm and the complexity analysis. We also updated several presentation issues.

As the revised manuscript is constrained by the same page limit requirement, we put (iii) and (iv) of the revision in the Appendix (and they will be merged into the main text for the final version). Please see the updated “​​Supplementary Material” for the entire revision.
We will answer these questions each reviewer asks individually.

---

### Meta-Review · Area_Chair_5qAh · 2022-08-25

**Recommendation:** Accept
**Confidence:** Less certain

**Metareview:**

This paper presents AMP, a method to automatically find the optimal parallelization strategy for large model training on heterogeneous compute resources taking into account the model architectures and cluster setups. The method is based on a model to estimate the cost of candidate strategies. A combination of heuristics and optimization techniques is used to determine the degree of parallelism, the micro batch size, the device assignment and the pipeline arrangement.

The reviewers agree that the paper addresses a relevant and timely challenge for model training, it proposes a solution that is more general than previous approaches, and it shows promising empirical performance. Most of the questions about the method and concerns about the presentation could be addressed in the rebuttal. The overall assessment of all reviewers is positive, I thus recommend acceptance, counting on the authors to take the feedback into account and incorporate the author response into the revision.

However, I want to emphasize that I think two concerns that came up are important and investing more in them could strengthen the paper a lot.
- First, the cost model is at the core of the contribution and underlies your parallelization strategy. So a more comprehensive ablation study (as requested by two reviewers) would be appropriate. The  spearman correlation numbers that you added in response provide valuable information, but they still leave many questions unanswered of how the cost model performs beyond the experimental setup you are currently using and how it behaves along different parameters in the cluster configuration. Such an ablation study could be evaluated independently of the benchmark results. Any effort along this direction to provide additional insights would be helpful and appreciated.
- Second, as you acknowledge experiments are at relatively small scale. However scalability is an important aspect of modern systems that is not discussed sufficiently in the paper. I understand that hardware resources might not be available and there is a justified use case also at smaller scale, but you should be forthcoming about the potential limitations of your model and the search algorithm that come with the size of the cluster you apply your algorithm to. Being clear about the scope of the work helps the reader  judge the applicability of your solution in practical problems.

**Award:**

No

---

### Decision · Program_Chairs · 2022-09-14

Accept